

# A glycolysis-related signature to improve the current treatment and prognostic evaluation for breast cancer

Sijie Feng[1], Linwei Ning[2], Huizhen Zhang[1], Zhenhui Wang[1] and Yunkun Lu[3]

[1] School of Medicine, Henan Polytechnic University, Jiaozuo, China
[2] School of Life Science and Technology, Xinxiang Medical University, Xinxiang, China
[3] Sir Run Run Shaw Hospital, Zhejiang University, Hangzhou, China

Corresponding authors
Zhenhui Wang,
wangzhenhui1984@hpu.edu.cn
Yunkun Lu, kevenloo1@zju.edu.cn

## ABSTRACT

**Background:** As a heterogeneous malignancy, breast cancer (BRCA) shows high incidence and mortality. Discovering novel molecular markers and developing reliable prognostic models may improve the survival of BCRA.

**Methods:** The RNA-seq data of BRCA patients were collected from the training set The Cancer Genome Atlas (TCGA)-BRCA and validation set GSE20685 in the Gene Expression Omnibus (GEO) databases. The "GSVA" R package was used to calculate the glycolysis score for each patient, based on which all the patients were divided into different glycolysis groups. The "limma" package was employed to perform differentially expression genes (DEGs) analysis. Key signature genes were selected by performing un/multivariate and least absolute shrinkage and selection operator (LASSO) C regression and used to develop a RiskScore model. The ESTIMATE and MCP-Counter algorithms were used for quantifying immune infiltration level. The functions of the genes were validated using Western blot, colony formation, transwell and wound-healing assay.

**Results:** The glycolysis score and prognostic analysis showed that high glycolysis score was related to tumorigenesis pathway and a poor prognosis in BRCA as overactive glycolysis inhibited the normal functions of immune cells. Subsequently, we screened five key prognostic genes using the LASSO Cox regression analysis and used them to establish a RiskScore with a high classification efficiency. Based on the results of the RiskScore, it was found that patients in the high-risk group had significantly unfavorable immune infiltration and prognostic outcomes. A nomogram integrating the RiskScore could well predict the prognosis for BRCA patients. Knockdown of PSCA suppressed cell proliferation, invasion and migration of BRCA cells.

**Conclusion:** This study developed a glycolysis-related signature with five genes to distinguish between high-risk and low-risk BRCA patients. A nomogram developed on the basis of the RiskScore was reliable to predict BRCA survival. Our model provided clinical guidance for the treatment of BRCA patients.

## INTRODUCTION

As a frequently diagnosed female malignancy (*Siegel et al., 2021*; *Shang & Xu, 2022*), breast cancer (BRCA) accounts for 15% (*Heer et al., 2020*) of all cancer-correlated deaths (*Nik-Zainal et al., 2016*). Invasive and metastatic characteristics (*Kim et al., 2018*) of BRCA are the leading causes of death to women aged 40 to 79 years old and males aged 60 to 79 years old (*Siegel et al., 2023*). The latest National Cancer Statistics Report showed that there were 297,790 and 59,910 new cases and deaths of BRCA in 2023, respectively (*Siegel et al., 2023*), exhibiting an obvious increasing trend as compared to 268,600 new cases and 41,760 death cases in 2019 (*Siegel, Miller & Jemal, 2019*). The regulating molecular mechanisms of the tumorigenesis and progression of BRCA are decisive to the heterogeneity of the cancer, which also poses great challenge to clinical treatment and disease prognosis for BRAC patients (*Lüönd, Tiede & Christofori, 2021*). High-throughput sequencing promotes the identification of molecular markers and development of effective treatment and intervention strategies for BRAC patients. This also suggests that developing reliable prognostic models can improve the overall survival (OS) of BRCA patients.

Metabolic reprogramming is an emerging hallmark of cancers. "Aerobic glycolysis" or the "Warburg effect" refers to the phenomenon that cancer cells favor glycolysis over mitochondrial oxidative phosphorylation in the breakdown of glucose even there is plentiful oxygen. A widely accepted explanation for this metabolic shift in cancer cells is that enhanced glycolysis is utilized by dividing cells as a flexible assembly line to produce glycolytic intermediates for multiple biosynthetic pathways to support rapid cell growth. In hepatocellular carcinoma, *Li et al. (2017)* demonstrated that genistein inhibits aerobic glycolysis through inactivating hypoxia inducible factor-1α (HIF-1α) to downregulate glucose transporter (GLUT1) and HexokinaseII. This suggested that developing glycolysis-related signatures may be a novel direction for cancer treatment and prognostic evaluation. *Zhang, Zhang & Yu (2019)* identified four glycolysis-related genes (DDIT4, AGRN, HMMR and AKR1A1) closely associated with the survival outcome of lung cancer patients. The glycolytic signature model established by *Wang et al. (2019)* can accurately predict the survival outcome of patients with endometrial cancer. This study contributed to the previous research on the role of glycolytic genes in BRCA by exploring the correlation between glycolysis and BRCA prognosis based on public databases. A prognostic model was developed with glycolysis-related genes and the correlation between risk score and immune infiltration landscape was analyzed. Furthermore, the impact of the key risk factors on the biological functions of BRCA cells was comprehensively explored. In conclusion, our study provided a novel direction for predicting clinical outcomes in BRCA patients.

## MATERIALS AND METHODS

### Data acquisition

The RNA sequencing (RNA-seq) data of 1082 BRCA and 114 para-cancer control samples were downloaded from the TCGA Genomic Data Commons (GDC) database (https://portal.gdc.cancer.gov/) and normalized by FPKM. The TCGA-BRCA expression profiles

were converted by log2 transformation. Subsequently, samples without survival status were removed to filter those with complete clinical information and survival times longer than 30 days but shorter than 10 years. Ultimately, a total of 994 BRCA samples were recruited for further studies. The expression data and clinical information of 327 tumor samples in GSE20685 were collected from the Gene Expression Omnibus (GEO) (https://www.ncbi.nlm.nih.gov/geo/) database. For the data from the GEO database, probes were mapped to genes by removing samples without survival status based on the annotation information of the corresponding microarray platform. The average value was taken as the expression value of the gene when multiple probes matched to a gene.

## Data preprocessing

A total of 994 BRCA samples with survival time >30 days and <10 years from the TCGA-BRCA were obtained, and the expression matrix of the RNA-seq data of these samples was transformed by log2. For GEO data, samples without survival information were eliminated. Based on the annotation information of chip platform, the mean expression value of a gene was taken when multiple probes matched to the gene.

## Tumor microenvironment analysis

ESTIMATE algorithm measures immune cell infiltration in the tumor microenvironment (TME) and could be used to explore the important role of stromal cells in tumor support and barrier function (Yoshihara et al., 2013). ESTIMATE algorithm was used in this study to compute immune and stromal scores in different risk groups from different cohorts. The MCP-counter algorithm estimates the absolute number of major cell types in the TME to evaluate the complexity of the TME in patients from different risk groups (Becht et al., 2016). The CIBERSORT algorithm analyzes the role of specific immune cell types in TME correlated with tumor prognosis and treatment response (Chen et al., 2018).

## Calculation of glycolysis score and DEGs

The hallmark_glycolysis gene set from Molecular Signatures Database (MSigDB, https://www.gsea-msigdb.org/gsea/msigdb/) (Liberzon et al., 2015) were downloaded to calculate glycolysis score using the "GSVA" R package (Hänzelmann, Castelo & Guinney, 2013). The optimum survival cutoff was determined applying the surv_cutpoint function in "survminer" package (Wang et al., 2021). The carcinogenic signaling pathway score was calculated using the "progeny" R package (Schubert et al., 2018). The DEGs between glycolysis score groups were filtered using "limma" package (Ritchie et al., 2015) (thresholds: adj.pval<0.05 and |log2 fold change|>log2(1.5)).

## Development of a glycolysis-related prognostic model

Key genes influencing the BRCA prognosis were selected from overlapping DEGs by univariate Cox regression analysis using the survival R package (Liu et al., 2021) (filter setting: $p$ value < 0.01). LASSO Cox regression with 10-fold cross-validation to progressively reduce the number of key genes in "glmnet" package (Wang et al., 2020). Subsequently, multivariate Cox regression analysis (Liu et al., 2021) was conducted to formulate a glycolysis-related RiskScore $= \sum(\text{cox coefficient} * \text{expression of gene})$.

Finally, the classification efficiency of the model was tested by the receiver operating characteristic (ROC) using "timeROC" package (*He et al., 2022*).

## Screening independent prognostic factors and development of a nomogram

Independent prognostic factors (*Zhang et al., 2021*) including the RiskScore, age and stage were selected by univariable and multivariable Cox regression analysis, with a $p < 0.05$ indicating a statistical significance. To better predict the prognosis of BRCA patients, a nomogram integrating the RiskScore, age and stage was developed using the "rms" package (*He et al., 2022*). The prediction accuracy and reliability of the nomogram were assessed by calibration curve and decision curve analysis (DCA) in "rms" R package (*He et al., 2022*).

## Immunotherapy and drug sensitivity analysis

Immunotherapy response could be evaluated immunophenoscore (IPS) (*Givechian et al., 2018*). The IPS of BRCA patients was collected from The Cancer Imaging Archive (TCIA, https://www.cancerimagingarchive.net/) database to identify sensitive BRCA responders to immunotherapy (*Prior et al., 2013*). Genes related to immune checkpoint inhibitor were collected from a previous study (*Hu et al., 2021*) to analyze the immunotherapy response in different risk groups. Subsequently, the "pRRophetic" R package was employed to predict drug sensitivity and calculate the correlation between RiskScore and half maximal inhibitory concentration (IC50) (*Geeleher, Cox & Huang, 2014*). Finally, drugs with an absolute value of relative coefficient >0.3 were selected.

## Cell culture and transfection

After testing mycoplasma contamination, the human BRCA cell line MCF7 (COBIOER Corp., Nanjing, China) was cultured in an incubator containing Dulbecco's modified Eagle's medium containing 100 μg/mL streptomycin and 100 U/mL penicillin (Invitrogen, Waltham, MA, USA) and 10% fetal bovine serum (FBS, Hyclone, Logan, UT, USA) in 5% $CO_2$ at 37 °C (*Akram, 2013*). Two specific small interfering RNAs (siRNA) were designed for PSCA silencing, and the regents (si-PSCA#1 and si-PSCA#1) was purchased from the Sigon corp. (Suzhou, China) and prepared to the required concentration following the manufacturer's protocol. The experiment was performed 48 h after the transfection.

## Western blot and transwell assay

After washing the cells with the ice-cold PBS, lysis buffer containing NaCl (125 mM), Tris-Cl (20 mM), 1 mM dithiothreitol (DTT), 0.1% Triton X-100, EDTA (0.5 mM) and protease inhibitor mixture was used to lysed the cells on ice for 15 min. Then, the cell lysate was ultra-sonicated for 10 seconds (s) three times and then incubated in cold condition for 45 minutes (s). After centrifugation (21,130 g) for 15 min, the supernatant was collected to measure the concentration using a Pierce BCA Protein assay kit (Waltham, MA, USA) following the manufacturer's protocol (*Akram, 2013*). Protein samples were isolated using a 10% sodium dodecyl sulfate-polyacrylamide gels and transferred to the nitrocellulose membrane, which was then blocked with 5% skimmed milk in TBST for 1 h before incubation with primary antibodies against PSCA overnight at 4 °C. Anti-β-actin antibody

served as an internal control. The membrane was incubated with conjugated secondary antibody horseradish peroxidase (HRP) for 1 h after washing with TBST three times. Finally, the Immobilon Western Chemiluminescent HRP substrate provided high sensitivity for enhanced chemiluminescent (ECL) detection, which was completed using a Luminescent Image Analyzer (Fujifilm Life Science, Cambridge, MA, USA). Transwell assay was carried out to assess cell invasion. A total of $3 \times 10^4$ cells in the upper chamber (containing 8-μm pore) in serum-free DMEM medium (100 μL) for 24 h, while the lower chamber contained 600 μL DMEM medium. Next, the migrated or invaded cells were fixed by 4% paraformaldehyde for 15 min and then stained by 0.1% crystal violet. The cells were photographed under an inverted microscope.

## Colony formation and wound healing assay

The MCF7 cells in the normal control (si-NC) and silenced groups (si-PSCA#1 and si-PSCA#2) were incubated in six-well plates (1,000 cells/well) containing DMEM medium with for 10 to 14 days. Subsequently, 4% paraformaldehyde and 0.1% crystal violet were used for cell counting (Akram, 2013). For wound-healing assay, the MCF7 cells were seeded into six-well plates and cultured until confluent. Next, a rectilinear scratch was generated applying a 100 μL pipette tip. The cells were similarly fixed and stained after incubation for 24 h. An inverted microscope was used for imaging the cells (Cappiello, Casciaro & Mangoni, 2018).

## Statistical analysis

All the statistical analyses and visualization were performed in R software (version 3.6.0). Significant differences in two sets of continuous variables were compared by the Wilcoxon rank-sum test. The spearman method was used for correlation analysis (Lin et al., 2015). KM survival analysis and log-rank test were used to compare the prognostic difference of patients ($p < 0.05$) (Liu et al., 2021). SangerBox (http://sangerbox.com/home.html) provided certain auxiliary analysis for this study.

# RESULTS

## Glycolysis was closely related to the progression of BRCA

The hallmark_glycolysis score was calculated for each tumor and para-cancer sample from TCGA, and it was found that tumor samples has higher glycolysis score (Fig. 1A). Based on the optimal survival cutoff of glycolysis score, the samples in the TCGA-BRCA and GSE20685 dataset were categorized into high- and low-glycolysis groups. The KM survival analysis revealed worse prognosis (Fig. 1B), disease free survival (DFS, Fig. 1C) and progress free survival (PFS, Fig. 1D) and poor prognosis (Fig. 1E) of patients in high-glycolysis groups. Moreover, carcinogenic signaling pathway score for each TCGA-BRCA patient was calculated. Analysis on the correlation between the glycolysis score and carcinogenic signaling pathway score showed that the glycolysis score was significantly positively correlated with multiple carcinogenic signaling pathways, especially the PI3 kinase Akt signaling (PI3K), epidermal growth factor receptor (EGFR), hypoxia, vascular endothelial growth factor (VEGF) pathway, and tumor necrosis factor-α (TNFα)

Peerj

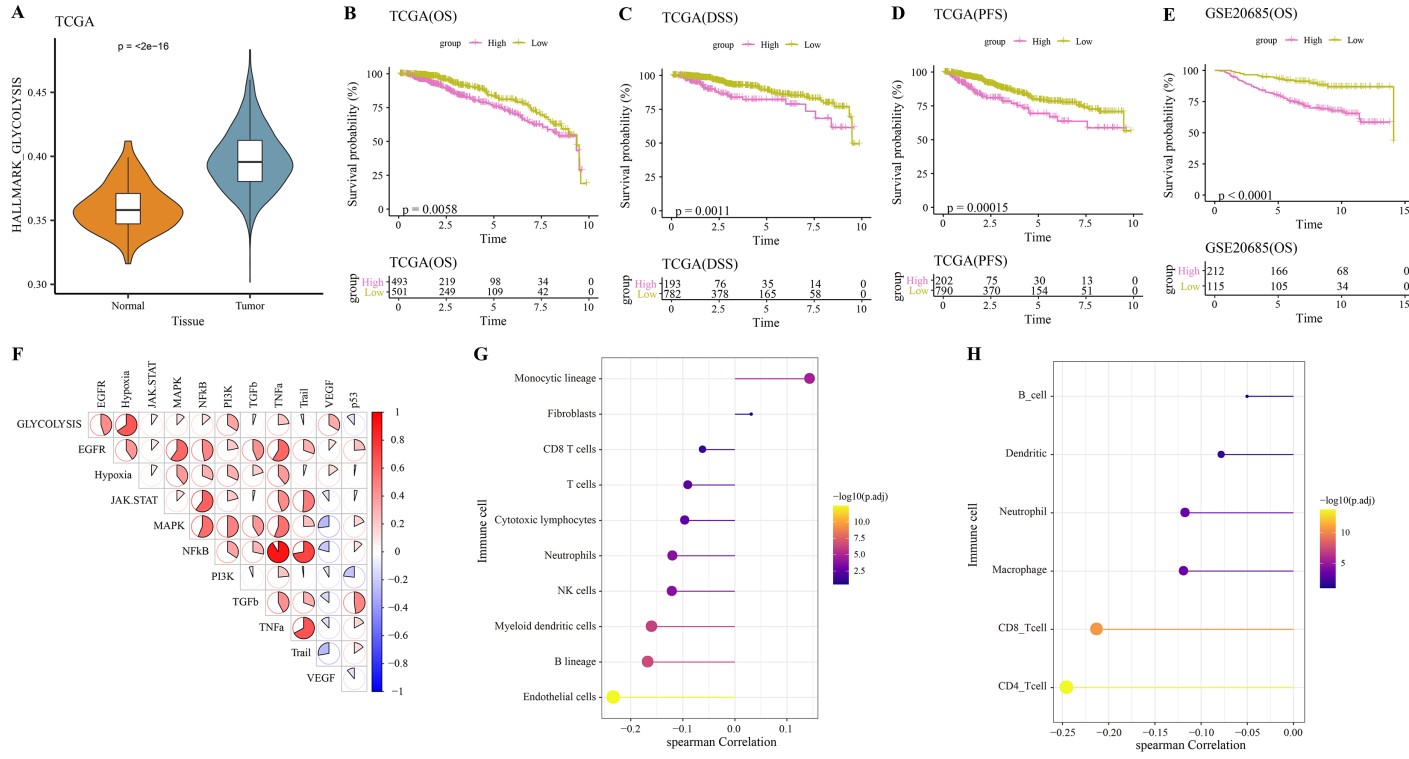

**Figure 1 Glycolysis score and tumor microenvironment analysis.** (A) Differences of glycolysis scores between tumor and paracancer samples in TCGA cohort. (B) Kaplan-Meier (KM) survival analysis of high-glycolysis scores group and low-glycolysis scores group in TCGA cohort. (C) Disease free survival (DFS) analysis of high-glycolysis scores group and low-glycolysis scores group in TCGA cohort. (D) Progress free survival (PFS) analysis of high-glycolysis scores group and low-glycolysis scores group in TCGA cohort. (E) Kaplan-Meier (KM) survival analysis of high-glycolysis scores group and low-glycolysis scores group in GSE20685 cohort. (F) The correlation analysis between the glycolysis score and the carcinogenic signaling pathway. (G) The correlation analysis between the glycolysis score and MCP-counter immune score. (H) The correlation analysis between the glycolysis score and TIMER immune score.

(Fig. 1F) and was significantly negatively correlated with the MCP-counter and TIMER immune scores of multiple immune cells such as the Neutrophils, NK cells and myeloid dendritic cells (Fig. 1G), macrophage, CD8_T cells and CD4_T cells (Fig. 1H). This indicated that overactive glycolysis could inhibit the functions and number of immune cells.

## Construction of a glycolysis-related risk model

The DEGs were identified between high-glycolysis and low-glycolysis groups in the TCGA-BRCA and GSE20685 datasets (Figs. 2A, 2B), and there were 299 overlapping genes (Fig. 2C). Pathway enrichment analysis revealed that these overlapping genes were mainly enriched in cell cycle, progesterone-mediated oocyte maturation, and oocyte meiosis (Fig. 2D), indicating that glycolysis promoted the proliferation of cancer cells. Next, univariate Cox regression and LASSO Cox regression analysis were performed to determine the key prognostic genes from these 299 genes (Fig. 3A). Multivariate Cox regression analysis eventually selected five prognostic genes for BRCA and their regression coefficient was calculated (Fig. 3B). The formula of the model was RiskScore $= (-0.102 * FREM1) + (-0.051 * NPY1R) + (-0.075 * CCL19) + 0.074 * PSCA + (-0.061 * CLIC6)$. Each patient was assigned with a RiskScore and the optimal survival

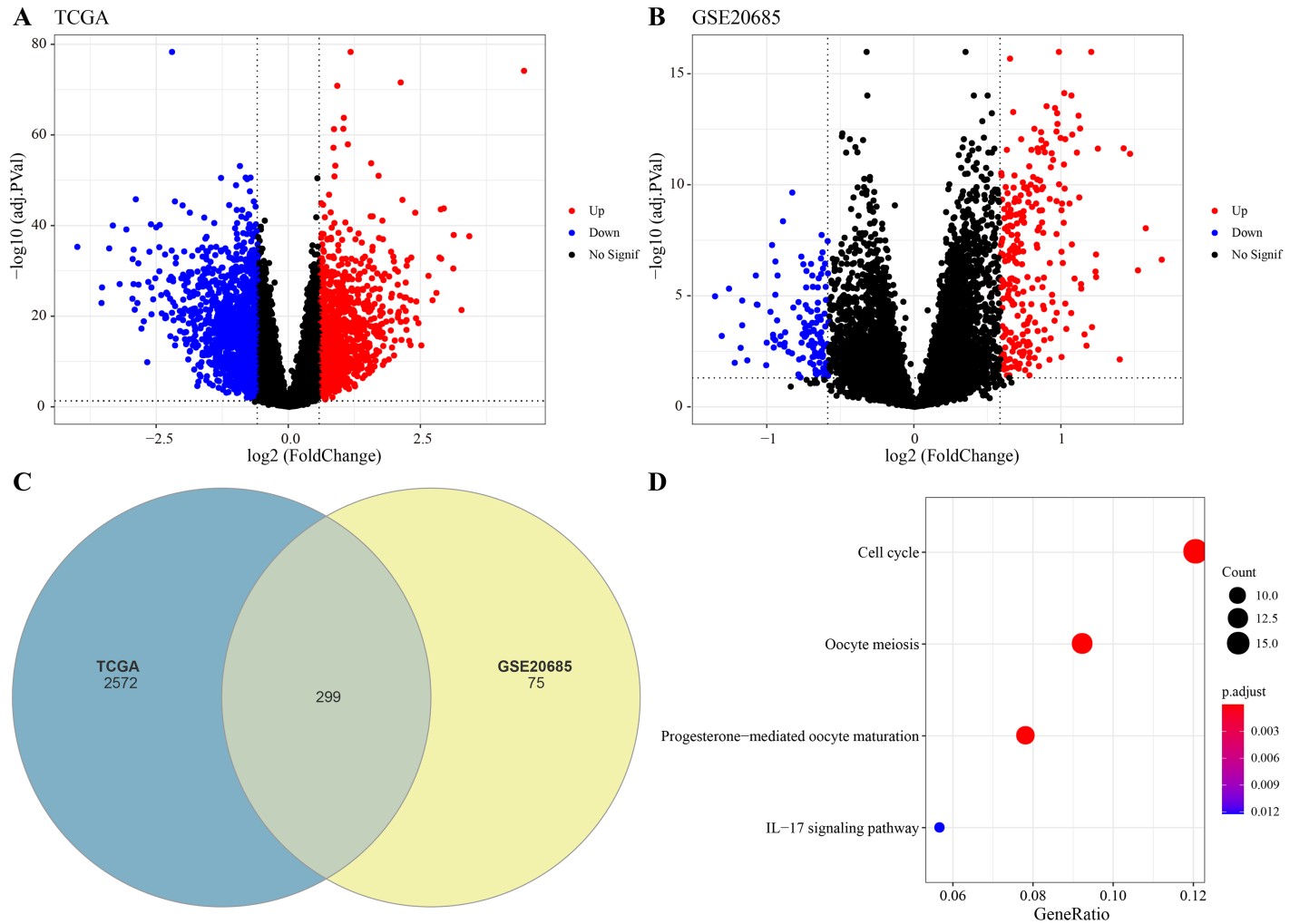

**Figure 2 Identification of differential expression genes (DEGs).** (A) Volcano plot of DEGs between the high-glycolysis scores group and low-glycolysis scores group in TCGA cohort. (B) Volcano plot of DEGs between the high-glycolysis scores group and low-glycolysis scores group in GSE20685 cohort. (C) Venn plot of overlapping genes between the TCGA and GSE20685 cohorts. (D) The enrichment pathway analysis of overlapping genes.

cutoff of the RiskScore could effectively distinguish between high- and low-risk patients. KM survival analysis revealed that the overall survival (OS) of high-risk patients were significantly worse ($p < 0.0001$, Fig. 3C). ROC analysis demonstrated that the AUC value of 1-, 3- and 5-year survival was 0.75, 0.71 and 0.65, respectively (Fig. 3D), showing a strong classification performance of the RiskScore. Comparison on the expression levels of the five model genes in different risk groups showed that only the high-risk patients had high-expressed PSCA (Fig. 3E), which was consistent with the results from multivariate Cox regression analysis as it showed that PSCA was a risk factor (Hazard ratio > 1 and $p < 0.02$). In addition, the OS of high-risk patients in the validation set (GSE20685) was significantly worse as compared to the low-risk patients (Fig. 3F), with 1-, 3- and 5-year AUC value of 0.75, 0.75 and 0.69, respectively (Fig. 3G). PSCA was high-expressed in

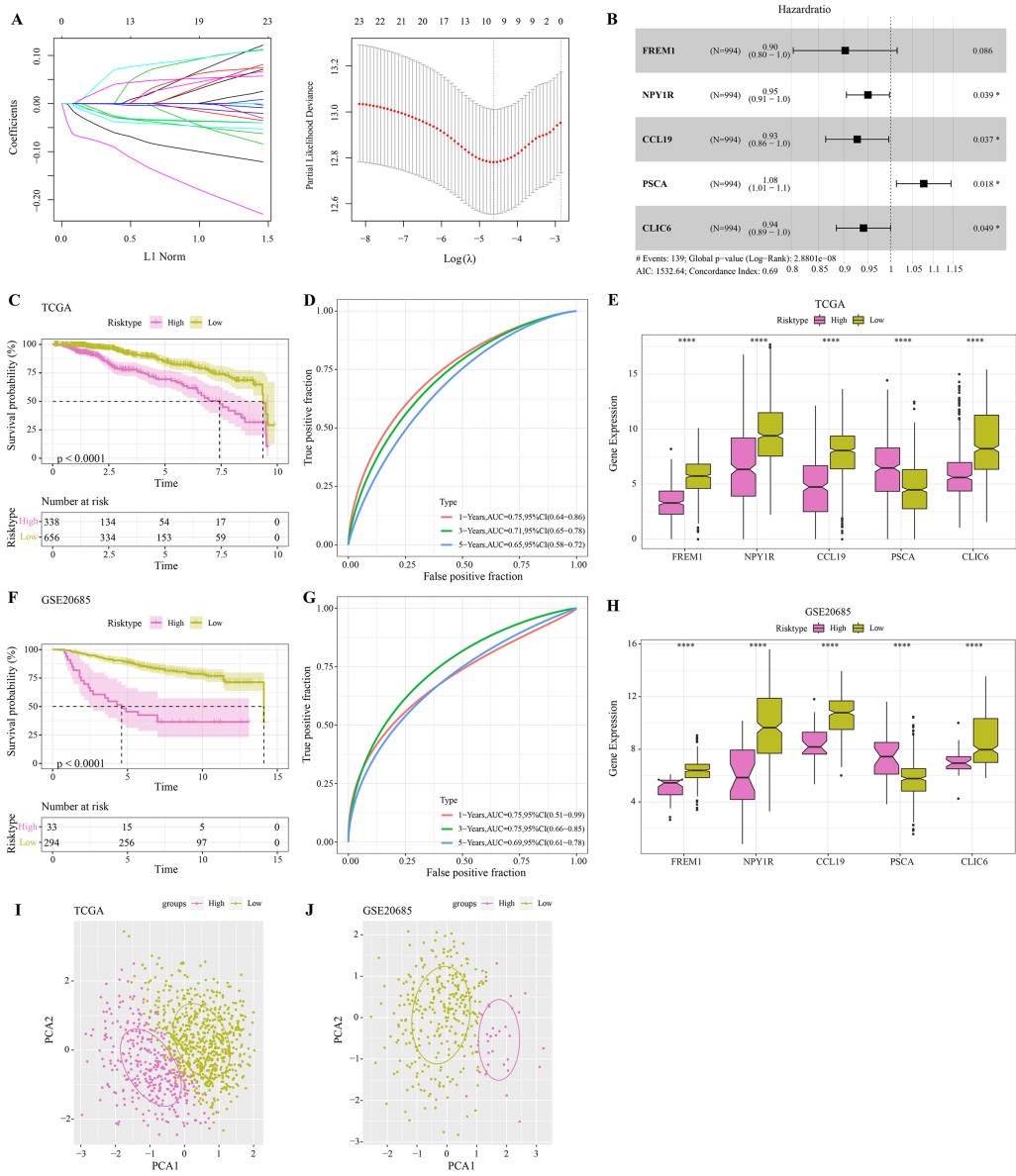

**Figure 3 Construction and validation of RiskScore model.** (A) Distribution of LASSO coefficients of the prognostic gene signature and the lambda confidence interval analysis. (B) Multivariate Cox regression of key prognostic genes. (C) High-risk group and low-risk group in TCGA cohort were subjected to KM survival analysis. (D) ROC analysis of 1-, 3- and 5-years prognosis in TCGA cohort. (E) The expression levels of five model genes in high-risk and low-risk groups in TCGA cohort. (F) KM survival analysis of high-risk group and low-risk group in GSE20685 cohort. (G) ROC analysis of 1-, 3- and 5-years prognosis in GSE20685 cohort. (H) The expression levels of five model genes in high-risk and low-risk groups in GSE20685 cohort. (I) Principal component analysis (PCA) of TCGA cohorts based on the expression of five model genes. (J) PCA of GSE20685 cohorts based on the expression of five model genes. *$P < 0.05$, ****$P < 0.0001$.

high-risk patients as a risk factor (Fig. 3H). Based on the expression of the five model genes, principal component analysis (PCA) showed that the five genes could effectively divide the patients in both TCGA and GSE20685 cohorts into two risk groups (Figs. 3I, 3J).

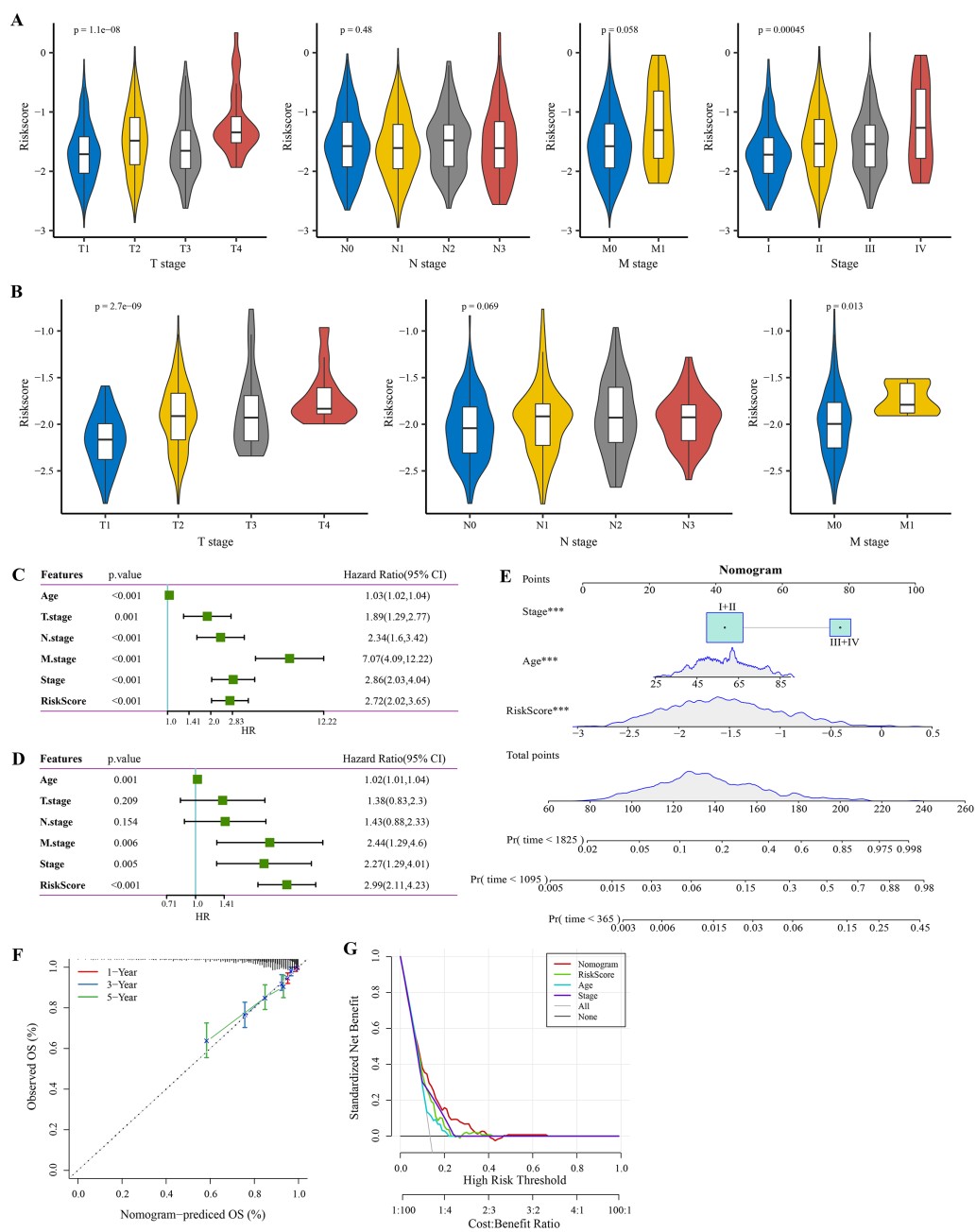

**Figure 4** **Independent factor determination and a nomogram developing.** (A) RiskScore difference of different clinical grades in TCGA cohort. (B) RiskScore difference of different clinical grades in GSE20685 cohort. (C) Univariate Cox regression analysis results shown in forest plot. (D) Multivariate Cox regression analysis results shown in forest plot. (E) Construction of a nomogram containing RiskScore, stage and age. (F) Calibration curve of nomogram. (G) Decision curve of nomogram.

## Screening independent factors and development of a nomogram

The RiskScore increased with lower clinical grade in the TCGA and GSE20685 cohorts (Figs. 4A, 4B). Univariate Cox regression analysis showed that the age, RiskScore, T. stage, N. stage, M. stage, and stage were significant prognostic factors for BRCA ($p < 0.001$,

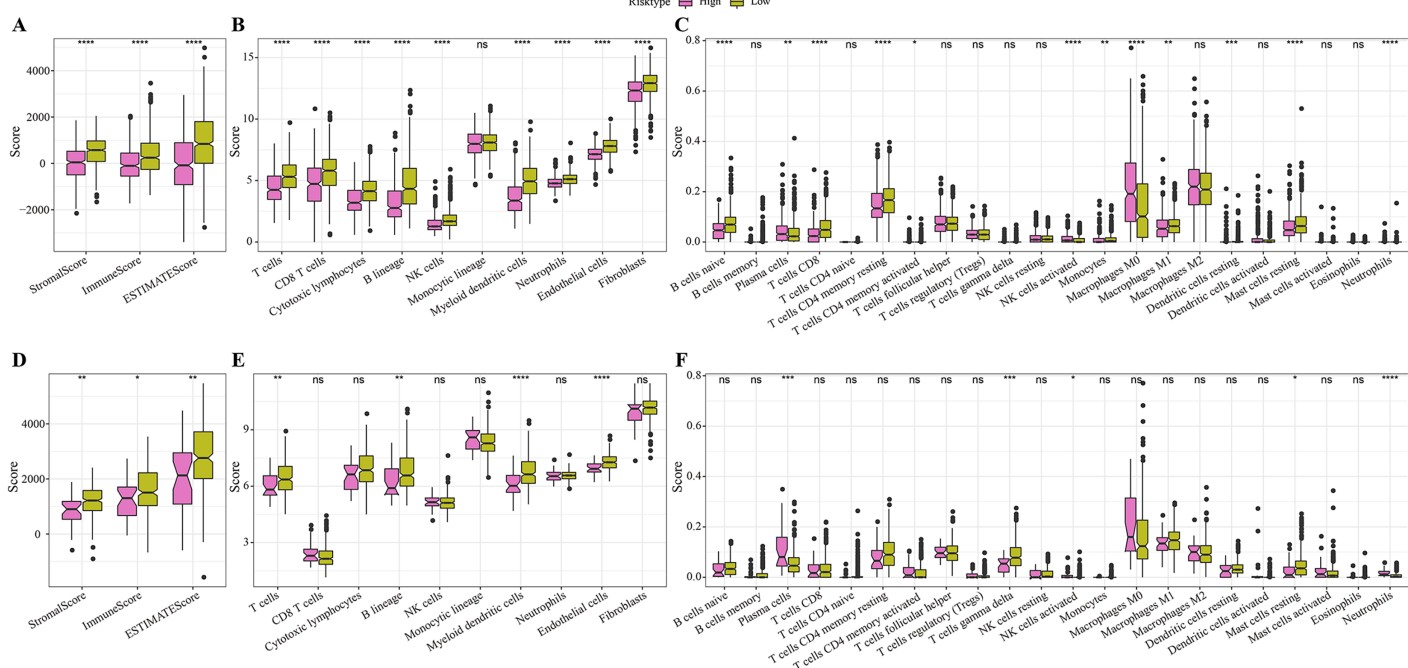

**Figure 5 Differences in immune microenvironment among different risk groups.** (A) ESTIMATE immune evaluation in TCGA cohort. (B) MCP-counter immune evaluation in TCGA cohort. (C) CIBERSORT immune evaluation in TCGA cohort. (D) ESTIMATE immune evaluation in GSE20685 cohort. (E) MCP-counter immune evaluation in GSE20685 cohort. (F) ESTIMATE immune evaluation in GSE20685 cohort. *$P < 0.05$, **$P < 0.01$, ***$P < 0.001$, ****$P < 0.0001$.

Fig. 4C). Multivariate Cox regression analysis further showed that the age, M. stage, stage, and RiskScore were significantly independent prognostic factors ($p < 0.001$, Fig. 4D). A nomogram model for predicting 1-, 3- and 5-year survival was developed (Fig. 4E). The calibration curves of the nomogram at 1-, 3- and 5-year points was highly consistent with the standard curves (Fig. 4F), indicating an accurate prediction of the nomogram. Also, the decision curve displayed a significantly higher benefit of the nomogram than the extreme curve (Fig. 4G), and the nomogram had strong prediction performance than other clinicopathological features.

## Immune infiltration differences among different risk groups

In the TCGA cohort, the low-risk group showed higher stromal score, immune score and ESRIMATE score (Fig. 5A). MCP-counter analysis demonstrated that this low-risk group was associated with the higher infiltration levels of immune cells, such as the CD8 T cells, NK cells, T cells, B lineage cells, endothelial cells, and myeloid dendritic cells (Fig. 5B). The results of CIBERSORT calculation revealed that the low-risk group had significantly more CD8+ T cells, M1 macrophages, NK cells activated, and mast cells (Fig. 5C). Moreover, the immune infiltration in the validation set (GSE20685) was also quantified. The results demonstrated that the low-risk samples were associated with higher stromal score and immune score (Fig. 5D) and had higher infiltration levels of lineage cells, myeloid dendritic cells, T cells, CD8 T cells, NK cells (Figs. 5E, 5F). These finding

suggested that the abundance of immune cells were significantly reduced in the high-risk group and their normal functions were also inhibited in TME.

## Immunotherapy and drug sensitivity analysis for different risk groups

The low-risk group with lower IPS were more likely to benefit from receiving immunotherapy treatment (Fig. 6A). Meanwhile, comparison on the antigen presentation of each risk group showed that the expression levels of related genes (such as the HLA-DPA1, HLA-DPB1 and HLA-DQA1) were higher in the low-risk group (Fig. 6B), while those of immune checkpoint inhibitor-related genes (such as the PDCD1, BTN2A2, VTCN1, ADORA2A, CTLA4 and TIGIT) were significantly higher in low-risk patients (Fig. 6C). Above these results indicated greater immunotherapy benefit for low-risk patients. The drug sensitivity analysis revealed that PHA-665752, DMOG, AZ628, AP-24534, Crizotinib, CP466722 and HG-6-64-1 were positively correlated with the low-risk group (Fig. 6D), and that pyrimethamine was negatively correlated with the low-risk group and had lower IC50 value in the high-risk patients (Figs. 6D, 6E). This suggested that pyrimethamine treatment could benefit high-risk patients more.

## PSCA was a key risk factors affecting the proliferation, invasion and migration of BRCA cells

Our model revealed that high-expressed PSCA was related to a higher RiskScore, and the function of PSCA was further verified using tumor cells. In the Western blot assay, the protein expression in the siPSCA#1 and siPSCA#2 groups was significantly downregulated ($p < 0.05$) as compared to the normal control group (Fig. 7A), indicating that the two independent small interfering RNAs had a high silencing efficiency. The transwell assay showed that the cell number was significantly reduced after silencing PSCA (Fig. 7B), suggesting a suppressed cell invasion activity. Additionally, in the colony formation assay, the number of cell colony numbers in the PSCA silencing groups were significantly less than that in the siNC groups (Fig. 7C), which indicated an inhibited cell proliferation ability. Meanwhile, the wound-healing assay showed that the cell migration ability was suppressed after PSCA silencing (Fig. 7D). These results validated that the PSCA was a key risk factors affecting the cell proliferation, invasion and migration in BRCA, and that high-expressed PSCA promoted BRCA progression.

## DISCUSSION

Recent findings in the studies of energy metabolism extend our traditional understanding of cancer as primarily a genetic disease to a metabolic disease (*Upadhyay et al., 2013*). The distinct metabolic characteristic of cancer cells, alternatively known as the "Warburg effect", means that cancer cells tend to favor glycolysis over oxidative phosphorylation even when oxygen is abundant (*Vaupel, Schmidberger & Mayer, 2019*). Emerging evidence indicates that the majority of cancer cells exhibit the "Warburg" metabolic phenotype, leading to a faster production of ATP compared to oxidative phosphorylation (*Graziano et al., 2017*; *Yan et al., 2017*; *Guo et al., 2021*). This study selected and analyzed genes related to glycolysis applying a series of bioinformatics analyses. Subsequently, a risk

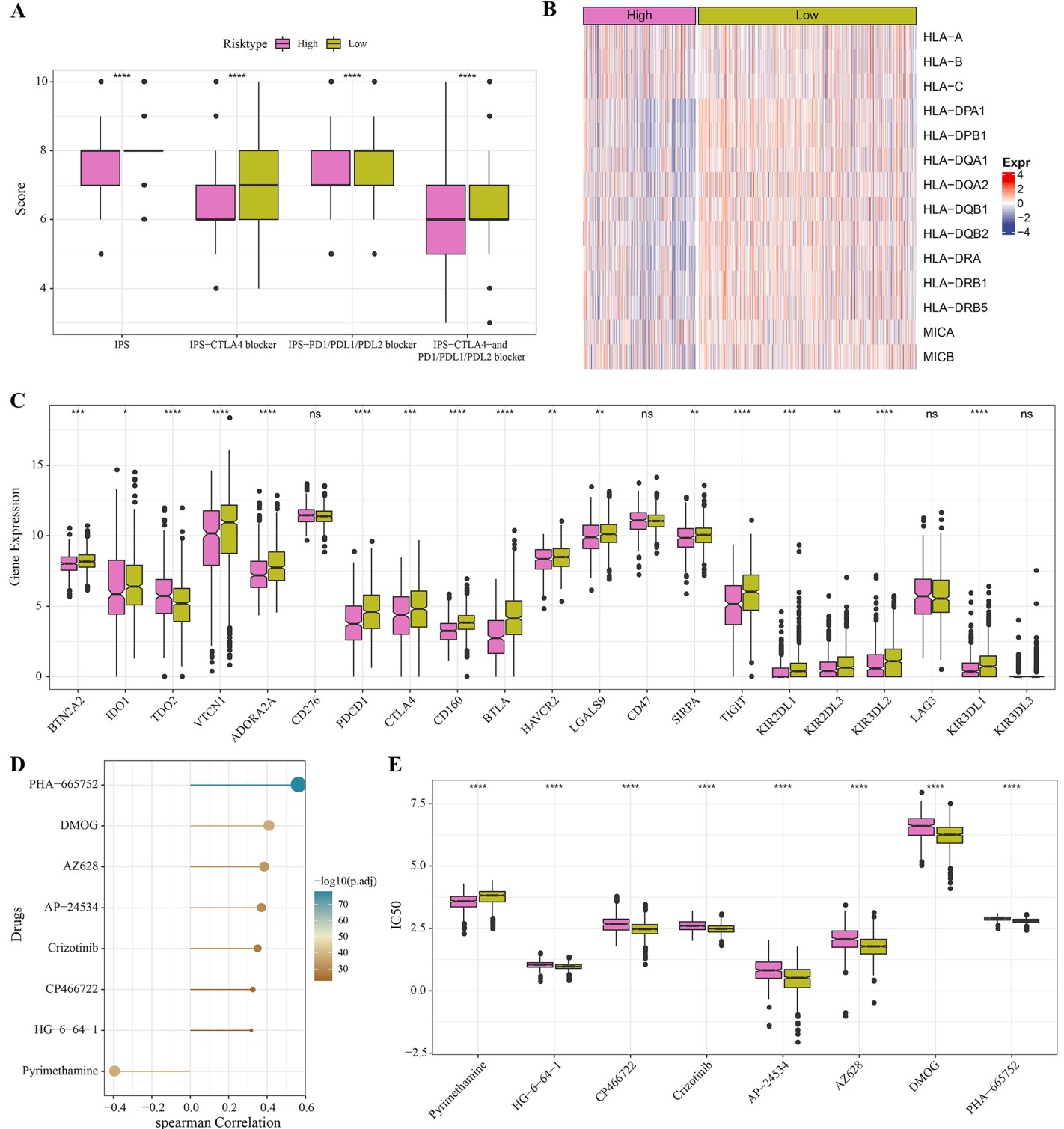

**Figure 6 Immunotherapy and drug sensitivity.** (A) IPS differences among different risk groups in TCGA cohort. (B) Differences in antigen presentation between different risk groups in TCGA cohort. (C) Differences in immune checkpoint inhibitor gene expression between different risk groups in TCGA cohort. (D) Correlation analysis between the RiskScore and drug sensitivity. (E) IC50 differences among different risk groups in TCGA cohort. *$P < 0.05$, **$P < 0.01$, ***$P < 0.001$, ****$P < 0.0001$.

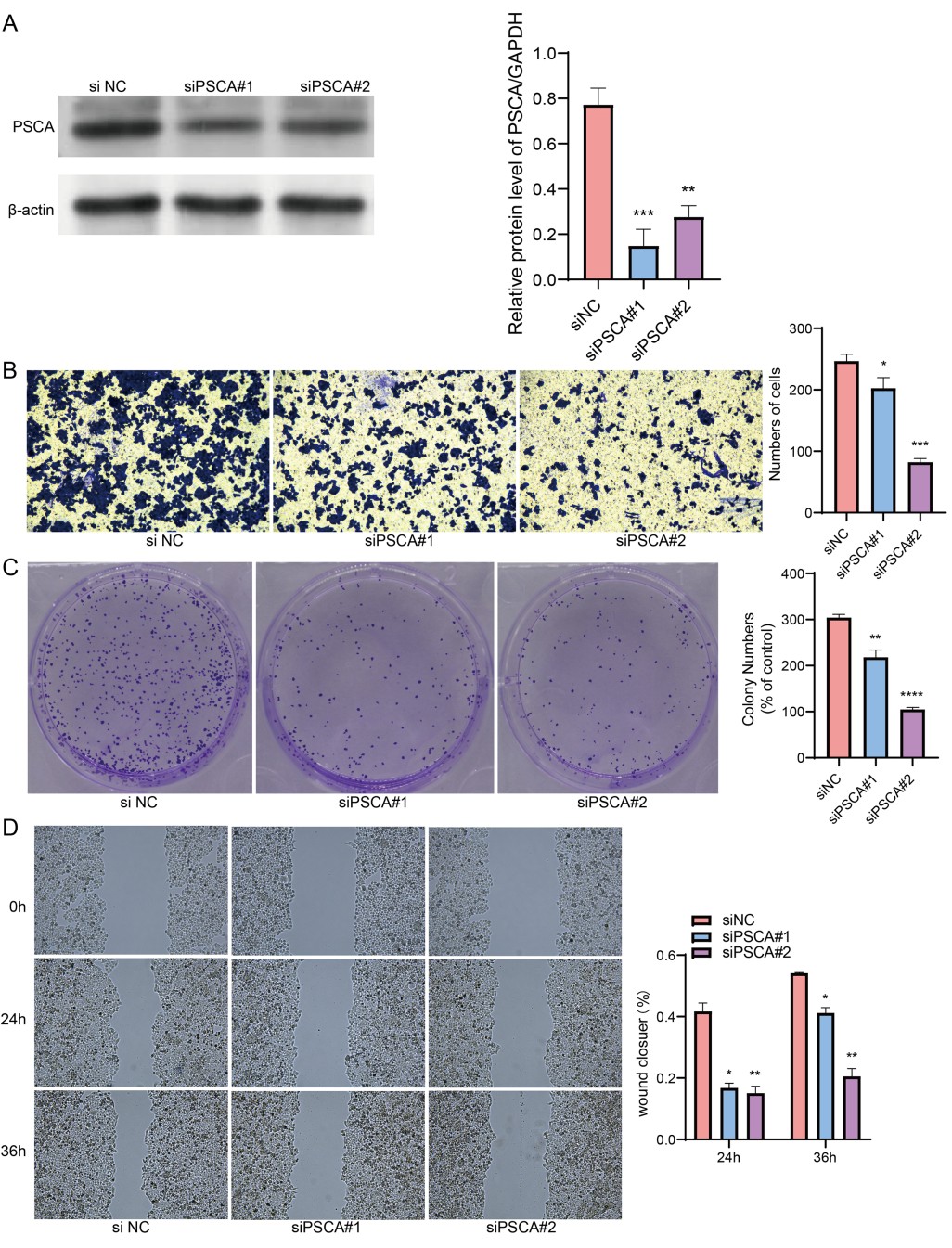

**Figure 7 Gene function identification *in vitro* experiment.** (A) Western blot assay for the expression test after PSCA silencing. (B) The trans-well assay for the cell invasion ability. (C) Colony formation assay for cell proliferation ability. (D) Wound healing assay for the cell migration ability. *$P < 0.05$, **$P < 0.01$, ***$P < 0.001$, ****$P < 0.0001$.

assessment model was developed utilizing multivariate and LASSO Cox regression analyses to select potential biomarkers for patients suffering from BRCA. The prognostic prediction of the model was highly accurate, highlighting its potential significance in the clinical diagnosis and management of BRCA patients.

In recent years, several studies have developed risk models for cancers using glycolysis-related genes. Based on TCGA and GEO databases and univariate Cox and LASSO regression analyses, *Guo et al. (2021)* constructed a risk model with five glycolysis-related genes to predict the prognosis for patients with prostate cancer. *Liu, Wang & Li (2022)* also screened genes related to glycolysis and immunity from public databases and developed a risk score model consisting of six glycolysis-immunity-related genes (ALDOC, VEGFA, PADI3, IGSF11, MIPOL1, and HGR) for oral squamous cell carcinoma. In addition, another previous study also screened six genes as independent prognostic genes for colon adenocarcinoma by performing enrichment analysis on the glycolysis-associated gene set but this model was not suitable for patients with rectal adenocarcinoma (*Liu et al., 2022*). These findings suggested a strong potential of using glycolysis-related genes as potential targets for cancer therapy.

This study identified PSCA as a risk gene (hazard ratio > 1) for BRCA, while FREM1, NPY1R, CCL19 and CLIC6 were the protective genes for the cancer (hazard ratio < 1). PSCA encodes cell membrane glycoprotein involved in signal transduction, cell adhesion and proliferation (*Wu et al., 2020*). This antigen has been reported as a critical marker in several cancers, such as prostate, bladder and pancreatic cancers (*Saeki et al., 2010*), and its high expression is related to a worse survival (*Kang et al., 2016*). A stable knockdown of PSCA can delay the growth, proliferation, metastasis and invasion of prostate cancer cells (*Liu et al., 2017*). In addition, the expression of PSCA is restricted to the tumor cells of BRCA and is upregulated in invasive BRCA but there is no clear association between PSCA-expression and treatment outcome, suggesting that PSCA cannot directly serve as a prognostic marker for patients with BRCA (*Link et al., 2017*). Consistently, this study also found that the PSCA was a key BRCA-promoting factor by enhancing the proliferation, invasion and migration ability of BRCA cancer cells, and that PSCA might be a potential treatment target. FREM1 encodes extracellular matrix (ECM) protein and plays a crucial part in the adhesion process of basement membrane and dermis (*Petrou, Makrygiannis & Chalepakis, 2008*). Previous study analyzed FREM1 expression profile with immunohistochemistry (IHC) and observed that a lower expression of FREM1 in BRCA is related to lower immune cell infiltration and worse overall survival (OS) (*Li et al., 2020*). Meanwhile, using various machine learning algorithms, *Xu et al. (2024)* screened FREM1 as one of the core genes for smog disease and found that the high FREM1 subgroup had significantly enriched glycosaminoglycan degradation and glycosphingolipid biosynthesis-ganglio series. NPY1R, an estrogen-responsive gene, is high-expressed in BRCA patients and mediates the inhibitory action to tumor cells. High expression of NPY1R as an independent prognostic factor is normally associated with higher endocrine sensitivity and better survival (*Bhat et al., 2022*). Based on *ex vivo* and *in vivo* studies, *Choong et al. (2021)* indicated that influencing islet peptide tyrosine tyrosine (PYY (1–36)) through NPY1R dependence improves glucose homeostasis in patients with type 2 diabetes. Chemokine CCL19 plays a pivotal role in tumor-suppressive functions (*Bhat et al., 2021*), for example, CCL19 promotes naïve T-cell survival *via* its anti-apoptotic activity (*Link et al., 2007*), enhances the endocytosis capacity of dendritic cells (DCs) (*Wu et al., 2023*), and induces the production of TNFα, IL-12 and IL-1β, indirectly leading to

increased proliferation of T cells and natural killer (NK) cells (*Laufer & Legler, 2018*). *Hayashi et al. (2021)* observed that overexpression of adipose-specific CCL19 can drive insulin resistance and weight gain in mice. In addition, CLIC6 is a channel protein involved in cell proliferation and homeostasis maintenance at early tumor stage, and its expression level is downregulated as malignant development progresses (*Shu et al., 2024*). However, the significance of CLIC6 in the glycolysis in cancer patients remained unknown, and our findings suggested that this gene was implicated in the glycolysis and progression of BRCA.

Overall, we constructed a RiskScore model for the prognostic prediction of BRCA based on the gycolytic metabolism characteristics. These key candidate genes in the RiskScore were associated with the adhesion, proliferation, homeostasis maintenance and immune cell protection process. High-risk BRCA patients exhibited lower immune infiltration and immunotherapy evasion, indicating that the immune microenvironment homeostasis was disrupted and recruited immune cells were reduced. For high-risk patients, pyrimethamine may be a more effective drug. Our study provided novel molecular targets for BRCA treatment but still had some limitations. First, the sample size of the dataset used in this study was relatively small. Therefore, it is important to collect more samples of BRCA patients, including those from different regions and ethnicities. In addition, further animal model experiments should be carried out to verify the role of the key genes in BRCA progression.

## CONCLUSIONS

The current study developed a reliable five-gene glycolysis-related prognostic model as an independent prognostic factor for BRCA. High-risk patients were closely correlated with lower immune infiltration, higher immunotherapy evasion and worse prognosis. In addition, a nomogram model integrating the RiskScore, age and stage can accurately predict both long- and short-term survival for BRCA patients with different RiskScores. PSCA was a key risk factor that affected the cell proliferation, invasion and migration of BRCA cells. It is hoped that the current model can become a reliable clinical tool to help clinicians make treatment decisions for BRCA patients with different risks.

## ABBREVIATIONS

| | |
|---|---|
| **BRCA** | Breast cancer |
| **GEO** | Gene Expression Omnibus |
| **TCGA** | The Cancer Genome Atlas |
| **DEGs** | Differentially expressed genes |
| **OS** | Overall survival |
| **TME** | Tumor microenvironment |
| **PDK** | Pyruvate dehydrogenase kinase |
| **GLUT** | Glucose transporters |
| **MSigDB** | Molecular Signatures Database |
| **LASSO** | Least Absolute Shrinkage and Selection Operator Regression |
| **ROC** | Receiver Operating Characteristic |

| IPS | Immuno-phenom-score |
|---|---|
| TCIA | The Cancer Imaging Archive |
| IC50 | Half maximal inhibitory concentration |
| KM | Kaplan-Meier |
| DFS | Disease free survival |
| PFS | Progress free survival |
| EGFR | Epidermal Growth Factor Receptor |
| PI3K | PI3 Kinase Akt Signaling |
| TNFα | Tumor necrosis factor-α |
| VEGF | Vascular endothelial growth factor |
| AUC | Area Under Curve |

### Funding

This study was supported by the Doctoral Fund of Henan Polytechnic University (No. B2020-51), the Science and Technology Project of Henan Province of China (No.242102310296) and the Fundamental Research Funds for the Universities of Henan Province (NSFRF240631). The funders had no role in study design, data collection and analysis, decision to publish, or preparation of the manuscript.

### Grant Disclosures

The following grant information was disclosed by the authors:
Doctoral Fund of Henan Polytechnic University: B2020-51.
Science and Technology Project of Henan Province of China: 242102310296.
Fundamental Research Funds for the Universities of Henan Province: NSFRF240631.

### Competing Interests

The authors declare that they have no competing interests.

### Author Contributions

- Sijie Feng conceived and designed the experiments, analyzed the data, prepared figures and/or tables, and approved the final draft.
- Linwei Ning conceived and designed the experiments, prepared figures and/or tables, authored or reviewed drafts of the article, and approved the final draft.
- Huizhen Zhang performed the experiments, authored or reviewed drafts of the article, and approved the final draft.
- Zhenhui Wang performed the experiments, prepared figures and/or tables, authored or reviewed drafts of the article, and approved the final draft.
- Yunkun Lu analyzed the data, authored or reviewed drafts of the article, and approved the final draft.

## Data Availability

The dataset generated and/or analyzed during the current study are available at GEO: GSE20685.

The raw data is available at GitHub and Zenodo:

- https://github.com/KevenLoo/My-New-Raw-Data.git.
- KevenLoo. (2024). KevenLoo/My-New-Raw-Data: Raw data (v1.1.0). Zenodo. https://doi.org/10.5281/zenodo.11078791.

## Supplemental Information

Supplemental information for this article can be found online at http://dx.doi.org/10.7717/peerj.17861#supplemental-information.

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
