# Peer review of "A glycolysis-related signature to improve the current treatment and prognostic evaluation for breast cancer"

_PeerJ, doi:10.7717/peerj.17861_

## Round 0.1 · original submission · Major Revisions

The reviewers found your research topic interesting and appreciated your data analysis methods. However, they raised several important concerns that require substantial revisions:

1. Lack of Comparison with Existing Models (Reviewer 1): Reviewer 1 questioned the novelty of your model and requested a comparison with similar risk prediction models to highlight its advantages.

2. Insufficient Discussion of Clinical Significance (Reviewer 1 & 3):
Reviewer 1 sought clarification on the model's implications for breast cancer treatment selection.
Reviewer 3 recommended discussing the model's limitations, such as generalizability to different populations or treatment subgroups, and suggesting future clinical trials.

3. Limited Exploration of Molecular Mechanisms (Reviewer 3): Reviewer 3 pointed out the lack of in-depth exploration of the molecular mechanisms by which the identified genes regulate the glycolysis pathway, impact the immune microenvironment, and contribute to breast cancer progression. The reviewer suggested adding mechanistic experiments or discussions.

4. Unclear Data Source and Preprocessing (Reviewer 3): Reviewer 3 requested clarification on how the TCGA-BRCA and GSE20685 datasets were accessed and whether standardized preprocessing steps were applied to ensure comparability.

5. Insufficient Comparison of Immune Infiltration Methods (Reviewer 3): Reviewer 3 recommended a detailed comparison of ESTIMATE and MCP-Counter, highlighting their individual contributions to understanding the tumor immune microenvironment.

Your manuscript requires substantial revisions to address the reviewers' concerns regarding the model's novelty, clinical significance, and underlying molecular mechanisms. Additionally, clearer descriptions of data processing and methodology are needed.

Reviewer 1 ·

Basic reporting

In present study, the authors developed a glycolysis-related signature with 5 genes to distinguish the high- and low-risk patients of breast cancer through TGCA and GEO database. Immune cell infiltration and prognosis in different risk groups were analyzed by authors and vitro validation experiments were conducted.
There are several questions about present manuscript.

1. Whether similar risk prediction models have been reported in other studies, please illustrate in the discussion.
2. Does the prediction model have guiding implications for the selection of treatment therapy for breast cancer?
3. The manuscript has multiple grammatical errors that require correction.

Experimental design

no comment.

Validity of the findings

Authors should conduct the internal validation of the prediction model.

Reviewer 2 ·

Basic reporting

The manuscript titled "A glycolysis-related signature to improve breast cancer treatment and prognostic evaluation" introduces a glycolysis-related signature aimed at enhancing current treatment methods and prognostic evaluation in breast cancer.

Experimental design

The study developed a nomogram to provide more accurate survival time estimations, potentially leading to advancements in individualized therapies.

Validity of the findings

The nomogram used in the study showed significantly higher accuracy compared to existing methods, as indicated by the decision curve analysis.
These findings have the potential to improve current treatment approaches and contribute to the development of personalized therapies for breast cancer patients.

Additional comments

Provide more detailed justification in the introduction, specifically expanding on the knowledge gap being addressed to enhance the manuscript's overall coherence and impact.

Reviewer 3 ·

Basic reporting

Please refer to Part Four

Experimental design

Please refer to Part Four

Validity of the findings

Please refer to Part Four

Additional comments

The study designed a prognostic model based on glycolytic-related genetic signatures to improve survival prediction and treatment guidance for patients with breast cancer (BRCA). The authors combined tumor microenvironment, glycolytic activity and molecular markers to provide a new perspective for understanding the heterogeneity of breast cancer, and established a prognostic risk scoring model based on 5 genes, which has important clinical significance. Using a large number of samples from TCGA and GEO databases for analysis, the reliability and wide applicability of the research results are ensured. Statistical tools such as "GSVA", "limma", "Lasso", as well as the ESTIMATE and MCP-Counter algorithms were used to conduct a comprehensive assessment of the tumor microenvironment. The method selection was appropriate, which enhanced the scientific rigor of the study. By constructing and validating the Riskscore model based on glycolytis-related genes, this model can effectively distinguish high-risk and low-risk patients, which is helpful to realize individualized treatment and prognosis of breast cancer. At the same time, the functional verification of PSCA gene mentioned in the study further proves the biological rationality of the model. The study describes in detail the data processing process, pre-processing steps, model building and validation processes, which enables other researchers to replicate and extend this study.
Although key prognostic genes have been identified, the molecular mechanisms of how these genes specifically regulate the glycolysis pathway, affect the immune microenvironment, and promote the progression of breast cancer have not been fully explored. It is suggested to add experiments or discussions on mechanism exploration.
While the study mentions the use of TCGA-BRCA and GSE20685 datasets, it would be beneficial to clarify if these datasets were accessed directly or through a specific platform (e.g., UCSC Xena, cBioPortal), and whether any data preprocessing steps were standardized across datasets to ensure comparability.
The use of ESTIMATE and MCP-Counter algorithms for immune infiltration assessment is commendable, but providing a detailed comparison between these methods and their individual contributions to the understanding of the tumor immune microenvironment would enrich the study.
The study proposes a nomogram integrating the RiskScore for clinical application, which is laudable. However, discussing potential limitations of the model, such as the generalizability to different ethnic populations or treatment subgroups, and suggestions for future clinical trials or prospective studies to validate the model in real-world settings, would be appropriate.

---

## Round 0.2 · accepted · Accept

All three reviewers are satisfied with the revisions made to your manuscript. I have also reviewed the changes and confirm that you have addressed all of the reviewers' comments. Thus, I am happy to accept your manuscript for publication. Congratulations on your work!

Reviewer 1 ·

Basic reporting

OK

Experimental design

OK

Validity of the findings

OK

Additional comments

OK

Reviewer 2 ·

Basic reporting

This manuscript presents a study on breast cancer (BRCA) prognosis using a 5-gene glycolysis-related prognostic model.
 The researchers developed a Riskscore model based on the expression levels of five genes and found that high-risk patients had worse overall survival. 
 They also developed a nomogram model that integrated the Riskscore, Age, and Stage to predict short- and long-term survival. 
 The study further investigated the immune infiltration differences between high- and low-risk groups and found that low-risk patients had higher immune cell infiltration. 
 Additionally, the study analyzed the potential benefits of immunotherapy and drug sensitivity for different risk groups. 
 The researchers identified PSCA as a key risk factor affecting the proliferation, invasion, and migration of BRCA cells. 
 Overall, this study provides insights into the prognostic prediction and potential treatment strategies for BRCA patients based on glycolysis-related genes.
 The glycolysis-related signature and Riskscore model developed in this study can guide clinical decision-making and improve the treatment and prognostic evaluation for BRCA patients.I recommend the manuscript to be accepted for the publication.

Experimental design

The authors have adjusted the experimental design and improved the flow of the manuscript according to the suggestions.

Validity of the findings

All the findings and results have been validated and improved the descriptions accordingly.

Reviewer 3 ·

Basic reporting

The study designed a prognostic model based on glycolytic-related genetic signatures to improve survival prediction and treatment guidance for patients with breast cancer (BRCA). The authors combined tumor microenvironment, glycolytic activity and molecular markers to provide a new perspective for understanding the heterogeneity of breast cancer, and established a prognostic risk scoring model based on 5 genes, which has important clinical significance. Using a large number of samples from TCGA and GEO databases for analysis, the reliability and wide applicability of the research results are ensured. Statistical tools such as "GSVA", "limma", "Lasso", as well as the ESTIMATE and MCP-Counter algorithms were used to conduct a comprehensive assessment of the tumor microenvironment. The method selection was appropriate, which enhanced the scientific rigor of the study. By constructing and validating the Riskscore model based on glycolytis-related genes, this model can effectively distinguish high-risk and low-risk patients, which is helpful to realize individualized treatment and prognosis of breast cancer. At the same time, the functional verification of PSCA gene mentioned in the study further proves the biological rationality of the model. The study describes in detail the data processing process, pre-processing steps, model building and validation processes, which enables other researchers to replicate and extend this study.

Experimental design

no comment

Validity of the findings

no comment